# Exploring the effect of sequential antibiotic exposure in resistant *Escherichia coli* causing urinary tract infections: a proof of principle study

Lisa Göpel,[1] Laura Kirchhoff,[1] Olivia Gopleac,[1] Leif Tüeffers,[1] Susanne Hauswaldt,[1] Sébastien Boutin,[1,2,3] Jan Rupp,[1,2] Dennis Nurjadi[1,2]

**ABSTRACT**  Antimicrobial resistance development, particularly in infections such as urinary tract infections (UTIs), is a global clinical concern. The objective of this study was to determine if sequential antibiotic exposure with ciprofloxacin and mecillinam can reduce the growth of resistant clinical *Escherichia coli* strains, thus improving the effectiveness of antibiotic therapy. Six *E. coli* isolates with heterogeneous resistance to ciprofloxacin and/or mecillinam obtained from patients with UTIs were exposed to one of the antibiotics (0.75 × minimum inhibitory concentration, MIC) for 1 h. This was followed by treatment with the second antibiotic at different concentrations (0.0375/0.075/0.375/0.75 × MIC). Continuous growth measurements were conducted in order to assess the impact of sequential exposure. One representative strain was selected for intact cell counting. In addition, a checkerboard assay was conducted to investigate the synergistic impact of ciprofloxacin and mecillinam, and genetic analyses were performed to identify the mechanisms of resistance for all isolates. The six *E. coli* strains were phylogenetically different, and none exhibited a synergistic effect for ciprofloxacin and mecillinam. Our data suggest that sequential exposure to mecillinam and ciprofloxacin appears to reduce the growth capacity of clinical *E. coli* isolates with phenotypic resistance to either or both agents. Sequential antibiotic exposure may be an interesting strategy to improve the antibiotic efficacy of current agents to overcome phenotypic resistance in *E. coli*.

**IMPORTANCE**  As global rates of antibiotic resistance increase and the development of new antibiotics become more difficult and costly, it is important to explore alternative strategies to improve the effectiveness of existing antibiotics. Previous studies have shown that sequential exposure of *Pseudomonas aeruginosa* to two antibiotics can effectively kill the bacteria and reduce the likelihood of resistance developing. However, the potential of this sequential approach for the treatment of *Escherichia coli* infections has not been thoroughly investigated. In our study, we conducted a proof-of-principle study to determine whether sequential exposure to mecillinam and ciprofloxacin can overcome phenotypic resistance to one or both drugs. We found that when *E. coli* were treated with subinhibitory doses of mecillinam followed by ciprofloxacin, their growth was significantly inhibited compared to treatment with ciprofloxacin alone, suggesting that sequential antibiotic exposure may be a viable strategy for treating infections caused by resistant *E. coli*.

**KEYWORDS**  sequential therapy, *E. coli*, UTI, mecillinam resistance, ciprofloxacin resistance

**Peer Reviewer** Balasubramaniyan Sakthivel, Anna University, Tiruchirapalli, India

Address correspondence to Dennis Nurjadi, dennis.nurjadi@uni-luebeck.de.

Jan Rupp and Dennis Nurjadi contributed equally to this article.

D.N. received speaker's honoraria from Shionogi and Cepheid. D.N. participated in advisory board of Shionogi outside the scope of this work.

See the funding table on p. 5.

*E*scherichia coli is the most common pathogen responsible for both uncomplicated and complicated urinary tract infections (UTIs), which are one of the most frequent bacterial infections worldwide (1). Increasing levels of antimicrobial resistance in uropathogenic *E. coli* have been reported. Due to the hurdles and challenges in developing new antimicrobial agents in both the pharmaceutical industry and academia, alternative methods and strategies for preserving the efficacy of existing antibiotics are needed to anticipate the problem of antibiotic resistance (2). One such strategy is to employ an evolution-informed therapeutic approach by exposing bacterial pathogens to two antibiotics in rapid succession. This evolution-informed approach to optimized bacterial treatment is based on the assumption that selective pressure from a given substance (with or without direct antimicrobial effects) can increase susceptibility to another antibiotic substance to which the organism is resistant in conventional antimicrobial susceptibility testing. Such an approach has been described by Roemhild *et al*. for *Pseudomonas aeruginosa* to improve antibiotic efficacy and reduce resistance selection (3). However, the use of sequential antibiotic exposure as a therapeutic strategy in resistant *E. coli* remains largely unexplored. In this study, we sought to test our hypothesis that sequential antibiotic exposure can achieve a similar effect in *E. coli*, thus providing experimental evidence as a proof of concept for further clinical validation.

An initial experiment with *E. coli* ATCC25922 and GM2163 was conducted to investigate the effects of sequential antibiotic combinations on the growth behavior of susceptible *E. coli*. The antibiotic panel was selected based on commonly prescribed antibiotics for UTI, including ciprofloxacin, fosfomycin, mecillinam, and trimethoprim-sulfamethoxazole. The experiments were performed using all substances against each other in both directions as a pretreatment and main treatment resulting in 12 potential combinations (the same substance for both pretreatment and main treatment was not performed) in each direction. As illustrated in Fig. S1, sequential exposure to first mecillinam followed by ciprofloxacin resulted in enhanced bacterial growth in ATCC25922, whereas bacterial growth was reduced under the reverse drug order. The GM2163 strain demonstrated the most pronounced effect on growth behaviour when exposed to mecillinam/ciprofloxacin or ciprofloxacin/mecillinam, in comparison with all other tested combinations (Fig. S1).

Based on these initial observations, we then repeated the assay with ciprofloxacin and mecillinam for 20 randomly selected UTI-causing *E. coli* strains from routine microbiological diagnostics, collected in March 2023 and June 2023. Of these, six strains were chosen based on their phenotypic resistance determined by broth microdilution in M9 medium to either ciprofloxacin (minimum inhibitory concentration, MIC >0.5 mg/L; Ecoli01-03) or mecillinam (MIC >8 mg/L; Ecoli04 and 05) or both (Ecoli06) (Table 1). Isolates were further characterized using short-read genome sequencing (Methodology, see Supplementary Material), and genomic analysis revealed that all isolates were phylogenetically diverse (Table 1). Ciprofloxacin resistance was associated with non-synonymous substitution in the *gyrA*, *parC*, and *parE* genes, and the mecillinam resistance was associated with $bla_{TEM-1}$. The presence of additional mecillinam resistance-encoding genes or mutations that have previously been identified as conferring resistance to mecillinam in *E. coli* was not observed (Data set S1). To rule out a synergistic effect of ciprofloxacin and mecillinam, a checkerboard assay was performed. None of the *E. coli* isolates tested in this study exhibited a synergistic effect for ciprofloxacin and mecillinam (Table 1).

The pretreatment duration was kept constant at 0.75 MIC for 1 h for all experiments. Based on the standard dosing intervals for pivmecillinam (200–400 mg thrice daily) and for ciprofloxacin (250–500 mg twice daily), we chose 8 h for mecillinam and 12 h for ciprofloxacin as a meaningful time course to study the growth behavior after the addition of the main treatment at 0.0375, 0.075, 0.375, and 0.75 MIC (Fig. S2). To quantify and compare the effect of sequential antibiotic exposure in *E. coli* strains, the growth

**TABLE 1** Ciprofloxacin and mecillinam susceptibility profiles of clinical *E. coli* isolates used in this study[a]

| ID | MLST | Phylogroup | Serogroup | Ciprofloxacin[b] | | Mecillinam[b] | | Synergy testing | |
|---|---|---|---|---|---|---|---|---|---|
| | | | | MIC (mg/L) | Int. | MIC (mg/L) | Int. | FICI | Int. |
| Ecoli01 | ST1193 | B2 | O75:H5 | 16 | R | 1 | S | 0.553 | n. i. |
| Ecoli02 | ST162 | B1 | O76:H27 | 8 | R | 0.5 | S | 0.651 | n. i. |
| Ecoli03 | ST131 | B2 | O25:H4 | 16 | R | 1 | S | 0.999 | n. i. |
| Ecoli04 | ST127 | B2 | O6:H31 | 0.015 | S | 64 | R | 0.874 | n. i. |
| Ecoli05 | ST453 | B1 | O23:H16 | 0.25 | S | 16 | R | 0.842 | n. i. |
| Ecoli06 | ST4981 | A | O8:H17 | 32 | R | 32 | R | 0.782 | i. |

[a]Abbreviations: FICI = fractional inhibitory concentration index; Int. = Interpretation; MIC = minimal inhibitory concentration; MLST = multi-locus sequence type; n. i. = no interaction.
[b]Antibiotic susceptibility testing interpretation based on EUCAST clinical breakpoints version 14.0; S = susceptible, R = resistant.

curve data were analyzed using the Growthcurver package for R (Methodology, see Supplemental material). The area under the curve was calculated for each growth curve, and the difference in AUC ($\Delta$AUC) between the sequential exposure and single antibiotic exposure was determined in order to summarize and compare the effect on bacterial growth (Fig. 1).

The exposure of mecillinam, followed by ciprofloxacin, was able to reduce the growth capacity of a ciprofloxacin-resistant and mecillinam-resistant strain Ecoli06 compared with single exposure ciprofloxacin in a dose-dependent manner (Fig. 1a). Determination of viable cells by impedance flow cytometry (BactoBox, SBT Instruments; Methodology, see Supplemental material) showed a reduction in viable bacterial cells under sequential exposure (both ciprofloxacin/mecillinam and mecillinam/ciprofloxacin combination) compared to single exposure after 4 h of incubation (Fig. 1c). Overall, exposure to mecillinam prior to ciprofloxacin exposure was able to reduce the growth capacity in 3 of 6 (50%) *E. coli* strains (Ecoli04, Ecoli05, and Ecoli06) in a dose-dependent manner compared to the exposure to ciprofloxacin alone. The exposure to ciprofloxacin was able to reduce the growth capacity in only two strains (Ecoli02 and Ecoli06) without dose dependency.

Our *in vitro* data suggested that fast sequential exposure to subinhibitory concentrations of mecillinam and ciprofloxacin on resistant clinical *E. coli* isolates obtained from patients with UTIs could reduce the bacterial growth of three (50%) mecillinam-resistant *E. coli* strains. Interestingly, the growth of bacteria was reduced only during the first 6–10 h of ciprofloxacin exposure when pre-exposed to mecillinam. The concept of sequential therapy is not completely novel and has been shown to minimize the adaption rate and inhibit the evolution of multi-drug resistance in laboratory experiments (4, 5). Recent studies suggested that sequential therapy may be more efficient than monotherapy regimens by exploiting evolutionary trade-offs, such as collateral sensitivity. This phenomenon describes the increased susceptibility to one drug due to the acquisition of resistance to another drug and has been observed in various bacterial species, including *E. coli* (6, 7). In 2018, it was reported that mutated mecillinam-resistant *E. coli* strains isolated from patients with UTIs were more likely to have collateral sensitivity toward antibiotic drugs compared with other mutated resistant strains. In contrast, mutated ciprofloxacin-resistant *E. coli* showed cross-resistance towards other antibiotics, such as chloramphenicol, ceftazidime, and amoxicillin (8). These observations suggest that the phenotypic susceptibility could be modulated by combining different antimicrobial drugs. The identification of robust evolutionary trade-offs may facilitate the identification of drug combinations that are more efficient when used sequentially than when administered individually (9).

Our study has limitations as it is based solely on *in vitro* observations and focuses only on sequential exposure of two antibiotic substances on a small number of clinical *E. coli* isolates. We did not include other combinations of antibiotics for sequential therapy or test other pathogenic agents causing UTIs in our study design. The primary readout in our experimental setup was optical density, which was chosen as a proxy to enable

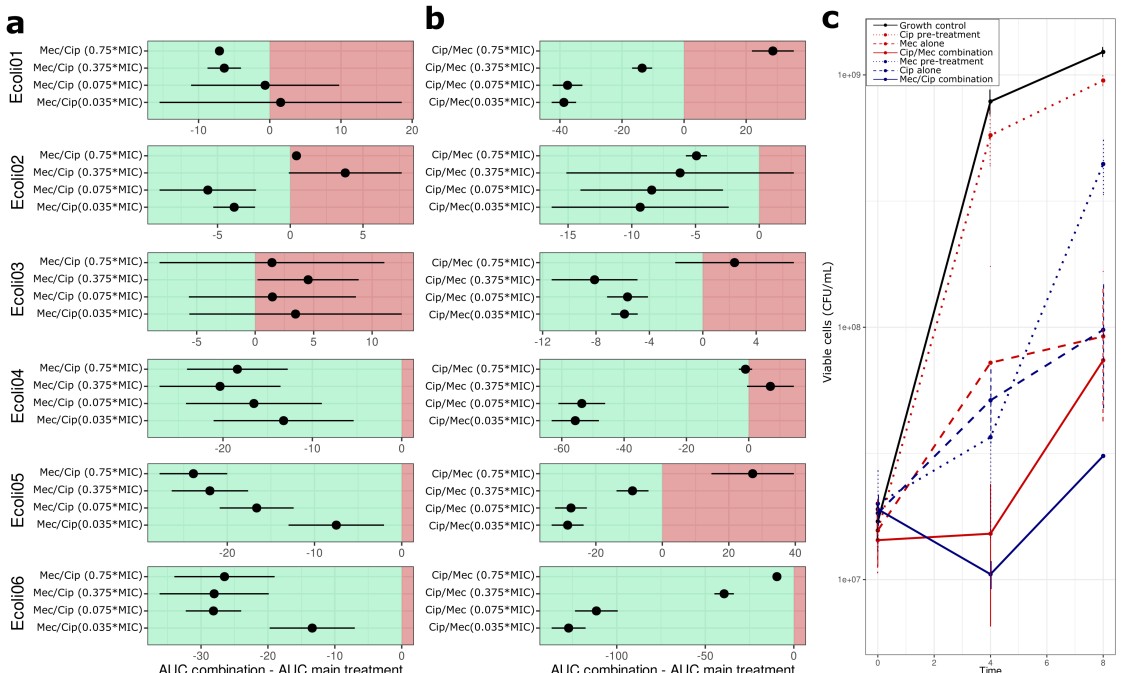

**FIG 1** Area under the curve (AUC) analysis of sequential antibiotic exposure for six *E. coli* strains and BactoBox measurement for one selected *E. coli* strain. (a) Isolates were incubated with mecillinam (0.75 × MIC) for 1 h, followed by ciprofloxacin exposure (0.0375 × MIC; 0.075 × MIC; 0.375 × MIC; 0.75 × MIC) for 12 h. The ΔAUC was calculated for sequential exposure with antibiotics in comparison with the AUC of exposure with ciprofloxacin alone. While Ecoli01–03 exhibited considerable variation in growth, Ecoli04–06 showed an overall decrease in growth. (b) Isolates were incubated with ciprofloxacin (0.75 × MIC) for 1 h, followed by mecillinam treatment (0.0375 × MIC; 0.075 × MIC; 0.375 × MIC; 0.75 × MIC) for 8 h. The ΔAUC was calculated for sequential exposure with antibiotics with the AUC of exposure with mecillinam alone. Only Ecoli06 exhibited reduced growth under sequential exposure in all used antibiotic concentrations compared to single exposure. (c) Viable cells were quantified over an 8-h period for E. coli06 using impedance flow cytometry. A reduction in the number of viable bacteria cells was observed following sequential exposure compared with single exposure, with a notable decline occurring after 4 h. The changes in the number of viable cells at timepoints 4 h and 8 h following combined treatment were not statistically significant when compared to those observed following single exposure (*P*-value = 0.7 for CIP_MEC exposure, *P*-value = 0.12 for MEC_CIP exposure).

higher throughput. This approach allowed us to avoid the time-consuming nature of traditional culture and cell counting methods, thereby increasing experimental capacity. However, additional measurements beyond optical density were also performed for one strain, as shown in Fig. 1c. Despite these limitations, our data suggested that sequential antibiotic exposure can reduce the growth capacity of clinical *E. coli* isolate even in isolates with phenotypic resistance to either or both substances. Thus, sequential antibiotic treatment may be a promising strategy for treating infections caused by resistant *E. coli* and warrants further investigation.

## ACKNOWLEDGMENTS

We are grateful for financial support from the German Science Foundation within the Excellence cluster Precision Medicine in chronic Inflammation (PMI, EXC 2167-390884018 to J.R.) and the BMBF within the German Center for Infection Research (DZIF, TTU 08.824 to J.R.). The authors thank Melanie Albrecht for her excellent technical assistance.

Study concept: L.G., S.B., J.R., D.N.; Experiments: L.G., O.G., L.K.; Bioinformatic analysis: S.B.; Statistical analysis and visualization: S.B.; Supervision: J.R., D.N.; Drafting: L.G., L.T., S.H., S.B., J.R., D.N.. Finalization of manuscript: all authors

## AUTHOR AFFILIATIONS

[1]University of Lübeck and University Hospital Schleswig-Holstein Campus Lübeck, Institute of Medical Microbiology and Infectious Diseases Clinic, Lübeck, Germany
[2]German Center for Infection Research (DZIF), Hamburg-Lübeck-Borstel-Riems, Lübeck, Germany
[3]Airway Research Center North (ARCN), German Center for Lung Research (DZL), Lübeck, Germany

## AUTHOR ORCIDs

Jan Rupp ⓘ http://orcid.org/0000-0001-8722-1233
Dennis Nurjadi ⓘ http://orcid.org/0000-0002-1278-5939

## FUNDING

| Funder | Grant(s) | Author(s) |
| --- | --- | --- |
| Deutsche Forschungsgemeinschaft (DFG) | EXC 2167-390884018 | Jan Rupp |
| Bundesministerium für Bildung und Forschung (BMBF) | TTU 08.824 | Jan Rupp |

## AUTHOR CONTRIBUTIONS

Lisa Göpel, Conceptualization, Writing – original draft, Writing – original draft, Writing – review and editing | Leif Tüeffers, Methodology, Writing – original draft, Writing – review and editing | Susanne Hauswaldt, Methodology, Writing – original draft, Writing – review and editing | Sébastien Boutin, Conceptualization, Methodology, visualization, Writing – original draft, Writing – review and editing | Jan Rupp, Conceptualization, Supervision, Writing – original draft, Writing – review and editing | Dennis Nurjadi, Conceptualization, Methodology, Supervision, Writing – original draft, Writing – review and editing.

## ADDITIONAL FILES

The following material is available online.

### Supplemental Material

**Data set S1 (Spectrum02525-2-S0001.xlsx).** Data on sequencing.
**Supplemental material (Spectrum02525-24-S0002.docx).** Supplemental methods; Figures S1 and S2.

### Open Peer Review

**PEER REVIEW HISTORY (review-history.pdf).** An accounting of the reviewer comments and feedback.

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
