## [Reviewer comments · Microbiology Spectrum]

Microbiology Spectrum

Exploring the effect of sequential antibiotic exposure in resistant *E. coli* causing urinary tract infections: A proof of principle study

Lisa Göpel, Laura Kirchhoff, Olivia Gopleac, Leif Tueffers, Susanne Hauswaldt, Sébastien Boutin, Jan Rupp, and Dennis Nurjadi

Corresponding Author(s): Dennis Nurjadi, Universitat zu Lubeck

Review Timeline:

Submission Date:	October 10, 2024
Editorial Decision:	December 10, 2024
Revision Received:	December 20, 2024
Accepted:	December 31, 2024

Editor: Aude Ferran

Reviewer(s): Disclosure of reviewer identity is with reference to reviewer comments included in decision letter(s). The following individuals involved in review of your submission have agreed to reveal their identity: Balasubramanian Sakthivel (Reviewer #1)

Transaction Report:

DOI: <https://doi.org/10.1128/spectrum.02525-24>

Re: Spectrum02525-24 (Exploring the effect of sequential antibiotic exposure in resistant E. coli causing urinary tract infections: A proof of principle study)

Dear Prof. Dennis Nurjadi:

Thank you for the privilege of reviewing your work. Below you will find my comments, instructions from the Spectrum editorial office, and the reviewer comments.

Revision Guidelines

Sincerely,
Aude Ferran
Editor
Microbiology Spectrum

Reviewer #1 (Comments for the Author):

Comments:

1. First, I thank all the authors involved in this research to overcome the urinary tract infections caused by resistant E. coli.
2. Ciprofloxacin and mecillinam drugs are used to treat urinary tract infections (UTIs) and they were resisted by E.coli due to the point mutations. These mutations were well identified in this study (p.S83L & p.D87N in GyrA and blaTEM-1 in AMR Gene).

3. These two drugs show a synergistic effect on resistant E. coli and it was reported in Table 1. If E. Coli (05 and 06) show resistance to both Ciprofloxacin and mecillinam, what are the alternate solutions to overcome the resistant E. coli?.

Reviewer #2 (Comments for the Author):

The paper describes an interesting study of sequential antibiotic exposure of ciprofloxacin and mecillinam against 6 strains of E.coli with various resistance mechanisms against the two drugs. The results are interesting in that there is effect measured as reduced growth of some of the strains after exposing first to one and then to the other drug.

Some comments:

1. Regarding the resistance mechanisms for mecillinam, it appears to me that the MIC levels for mecillinam against isolates 1-3 are such, that one would expect from strains with TEM1 - usually mecillinam MIC levels for susceptible strains are much lower (0.125); what then was the background for mecillinam resistance in strains 4-6? if only TEM1 was found, it would mean several copies of TEM1? A range of mutational resistance mechanisms is found for mecillinam.
2. What was the reason for using subMIC levels for the experiment? 0.75xMIC results for cipro resistant strains in concentrations that are far above those that can be achieved for cipro in serum after standard doses, and the same holds true for highly mecillinam resistant strains - these higher concentrations could be seen in urine, but are often much higher. So the experimental condition is rather far from the clinical situation.
3. SubMIC exposure for both drugs for extended period (as the longer time exposure here) would most probably lead to resistance mutations for both compounds - was this tested?
4. A cautionary note for using OD for growth is that bacterial cells expand in size under mecillinam exposure which may lead to false growth rates.
5. Supplementary figure 2 would benefit from using the same scale on the X-axis.

Review on Spectrum02525-24

Title: Exploring the effect of sequential antibiotic exposure in resistant *E. coli* causing urinary tract infections: A proof of principle study.

Comments:

1. First, I thank all the authors involved in this research to overcome the urinary tract infections caused by resistant *E. coli*.
2. Ciprofloxacin and mecillinam drugs are used to treat urinary tract infections (UTIs) and they were resisted by *E.coli* due to the point mutations. These mutations were well identified in this study (p.S83L & p.D87N in GyrA and blaTEM-1 in AMR Gene).
3. These two drugs show a synergistic effect on resistant *E. coli* and it was reported in Table 1. If *E. Coli* (05 and 06) show resistance to both Ciprofloxacin and mecillinam, what are the alternate solutions to overcome the resistant *E. coli*?

1 **Point-by-point response:**

2
3 We thank both reviewers for their encouraging and constructive comments. We hope to have addressed
4 all raised points in the revised manuscript (version R1).

5
6 **Reviewer #1 (Comments for the Author):**

7
8 Comments:

9 1. First, I thank all the authors involved in this research to overcome the urinary tract infections caused
10 by resistant *E. coli*.

11 2. Ciprofloxacin and mecillinam drugs are used to treat urinary tract infections (UTIs) and they were
12 resisted by *E. coli* due to the point mutations. These mutations were well identified in this study
13 (p.S83L & p.D87N in GyrA and blaTEM-1 in AMR Gene).

14 >>Response: Thank you for the encouraging comments.

15
16 3. These two drugs show a synergistic effect on resistant *E. coli* and it was reported in Table 1. If *E.*
17 *Coli* (05 and 06) show resistance to both Ciprofloxacin and mecillinam, what are the alternate
18 solutions to overcome the resistant *E. coli*?

19 >>Response: We appreciate your comment and have reviewed Table 1. We reported no synergistic
20 effect of all *E. coli* isolates tested for ciprofloxacin and mecillinam (as shown in Table 1, last column:
21 “no interaction”). However, by sequential antibiotic exposure of both substances to the ciprofloxacin
22 and mecillinam resistant strain *E. coli* 6, we observed a reduced growth capacity. This sequential
23 exposure of two substances to a strain that is phenotypically resistant to both substances could be a
24 promising strategy for the treatment of infections caused by resistant *E. coli*. At the time being, we do
25 not have any definitive explanation for the mechanism behind this phenomenon. One may hypothesize
26 that by exposing the bacteria for a short period of time to mecillinam (despite phenotypic resistance)
27 may induce slight damage to the cell wall, which then increase the vulnerability of the cell to a second
28 substance.

29
30
31 **Reviewer #2 (Comments for the Author):**

32
33 The paper describes an interesting study of sequential antibiotic exposure of ciprofloxacin and
34 mecillinam against 6 strains of *E. coli* with various resistance mechanisms against the two drugs. The
35 results are interesting in that there is effect measured as reduced growth of some of the strains after
36 exposing first to one and then to the other drug.

37
38 Some comments:

39 1. Regarding the resistance mechanisms for mecillinam, it appears to me that the MIC levels for
40 mecillinam against isolates 1-3 are such, that one would expect from strains with TEM1 - usually
41 mecillinam MIC levels for susceptible strains are much lower (0.125); what then was the background
42 for mecillinam resistance in strains 4-6? if only TEM1 was found, it would mean several copies of
43 TEM1? A range of mutational resistance mechanisms is found for mecillinam.

44 >>Response: Thank you for your helpful comment. Just as a clarification *E. coli* 1-3 does not carry
45 bla_{TEM-1} gene. As presented in the manuscript and the supplementary data, only the mecillinam
46 resistant strain (*E. coli* 4-6) carried the bla_{TEM-1}. We checked also the copy number and only one copy
47 of bla_{TEM-1} was found in every resistant isolate in our study (*E. coli* 4-6), while no other mecillinam
48 resistance-encoding genes or mutations previously associated with conferring mecillinam resistance in
49 *E. coli* were identified. The information about the genetic mechanisms of resistance were included in
50 the supplementary dataset but we have now also added one sentence in line 96 to 98.

51
52 2. What was the reason for using subMIC levels for the experiment? 0.75xMIC results for cipro
53 resistant strains in concentrations that are far above those that can be achieved for cipro in serum after
54 standard doses, and the same holds true for highly mecillinam resistant strains - these higher
55 concentrations could be seen in urine, but are often much higher. So the experimental condition is
56 rather far from the clinical situation.

57 >>**Response:** Since this study was conducted as a proof-of-principle, we specifically chose a
58 subinhibitory MIC. We acknowledge that our experimental setup is not perfect and may not fully
59 reflect in vivo conditions or the clinical setting. Further studies are necessary to translate our findings
60 into clinical practice.

61 At antibiotic concentrations \geq MIC, bacterial growth is inhibited, as expected. This makes it impossible
62 to study potential changes in the growth behavior or capacity of the *E. coli* strains following exposure.
63 For this reason, we utilized the sub-MIC approach, which is a conventional experimental setup widely
64 used to study resistance evolution in vitro (PMID: 30209218).

65
66
67

68 3. SubMIC exposure for both drugs for extended period (as the longer time exposure here) would most
69 probably lead to resistance mutations for both compounds - was this tested?

70 >>**Response:** Yes, we have performed serial passaging experiments using two approaches in the past:
71 doubling the concentration of ciprofloxacin per passage and maintaining a constant concentration per
72 passage. Our findings indicate that doubling the concentration selects for resistance more rapidly,
73 within five days, compared to constant exposure. Given that the typical duration of antibiotic therapy
74 for urinary tract infections (UTIs) is relatively short (typically 7 days or less, as noted in PMID:
75 34313686, 39495518), we believe that in vitro resistance experiments involving constant sub-MIC
76 antibiotic concentrations, which may take more than a week, are maybe less meaningful in this
77 context.

78

79 For our experiments, we chose sub-MIC exposure to study bacterial growth behavior following
80 sequential exposure. Using the MIC concentration resulted in bacterial killing, which was not suitable
81 for our research aim. The rationale for sub-MIC exposure was to observe whether growth capacity
82 could be reduced under such conditions, with the expectation that this effect would translate similarly
83 to MIC concentrations. To avoid any misunderstanding, we do not advocate the use of sub-MIC
84 concentrations of antibiotics in therapeutic settings.

85
86

87 4. A cautionary note for using OD for growth is that bacterial cells expand in size under mecillinam
88 exposure which may lead to false growth rates.

89 >>**Response:** Thank you for this comment. We also agree that OD measurements can only be used as
90 a proxy to estimate the growth behaviour of bacteria. This was the reason, why we performed
91 additional measurements, such as cell counting using the BactoBox as indicated on Figure 1 panel c.
92 We have added this into the limitation of the study.

93

94 5. Supplementary figure 2 would benefit from using the same scale on the X-axis.

95 >>**Response:** We appreciate your comment and have reviewed the scale of the X-axis in
96 Supplementary figure 2. We chose 12 hours for ciprofloxacin and eight hours for mecillinam as a
97 meaningful time course to study bacterial growth, based on the standard dosing interval for
98 pivmecillinam (200 - 400 mg three times daily/every eight hours) and for ciprofloxacin (250 – 500 mg
99 twice daily/every 12 hours) in the clinical setting. Given the structure of the growth curve experiments
100 performed, we retained the original scaling of the X-axis.

Re: Spectrum02525-24R1 (Exploring the effect of sequential antibiotic exposure in resistant *E. coli* causing urinary tract infections: A proof of principle study)

Dear Prof. Dennis Nurjadi:

Your manuscript has been accepted, and I am forwarding it to the ASM production staff for publication. Your paper will first be checked to make sure all elements meet the technical requirements. ASM staff will contact you if anything needs to be revised before copyediting and production can begin. Otherwise, you will be notified when your proofs are ready to be viewed.

Sincerely,
Aude Ferran
Editor
Microbiology Spectrum